# Sociodemographic Factors Affecting Older People’s Care Dependency in Their Daily Living Environment According to Care Dependency Scale (CDS)

**DOI:** 10.3390/healthcare9020114

**Published:** 2021-01-21

**Authors:** Grażyna Puto, Izabela Sowińska, Lucyna Ścisło, Elżbieta Walewska, Alicja Kamińska, Marta Muszalik

**Affiliations:** 1Department of Internal and Environmental Nursing, Institute of Nursing and Midwifery, Faculty of Health Sciences, Jagiellonian University Medical College, Kopernika Str. 25, 31-501 Krakow, Poland; izabela.sowinska@uj.edu.pl; 2Department of Clinical Nursing, Institute of Nursing and Midwifery, Faculty of Health Sciences, Jagiellonian University Medical College, Kopernika Str. 25, 31-501 Krakow, Poland; lucyna.scislo@uj.edu.pl (L.Ś.); elzbieta.walewska@uj.edu.pl (E.W.); 3Laboratory of Theory and Fundamentals of Nursing, Institute of Nursing and Midwifery, Faculty of Health Sciences, Jagiellonian University Medical College, ul. Michałowskiego 12, 31-126 Krakow, Poland; alicja.kaminska@uj.edu.pl; 4Department of Geriatrics, Collegium Medicum in Bydgoszcz, Nicolaus Copernicus, University in Torun, M.Skłodowskiej-Curie Str. 9, 67-090 Toruń, Poland; muszalik@cm.umk.pl

**Keywords:** older people, care dependency, living environment, care dependency scale (CDS)

## Abstract

The aim of the research was to determine the influence of sociodemographic factors on older people’s care dependency in their living environment according to the Care Dependency Scale (CDS). *Methods:* The research was conducted in a group of 151 older people staying in their own homes. The methods applied in the research included a sociodemographic questionnaire and scales including the Abbreviated Mental Test Score (AMTS), CDS, Katz Index of Independence in Activities of Daily Living (ADL), Lawton’s Instrumental Activities of Daily Living (I-ADL), Mini Nutritional Assessment (MNA), and Geriatric Depression Scale (GDS). *Results:* Gender had a significant impact on the level of care dependency. The surveyed females obtained the medium or high level of dependency more often than males (22.4% vs. 6.1%), and the low level of dependency was significantly more frequent among men than women (*p* = 0.006). Moreover, the age of the respondents determined their level of care dependency. The subjects with a medium or high level of care dependency were significantly older (*p* = 0.001). The subjects with a low level of care dependency were more likely to be married than people with a medium/high level (*p* < 0.001). The level of education had a significant impact on care dependency. A higher level of education correlated with a medium/high level of dependency (*p* = 0.003). *Conclusions:* The survey results confirmed that sociodemographic factors have a significant impact on the level of care dependency. When planning care in the home environment, special attention should be paid to older women, who are more likely to lose their independence than men. These women should be given additional support.

## 1. Introduction

Following the escalating phenomenon of an aging society in Poland and in other European countries, the problems connected with taking care of older patients have become particularly meaningful, not only from a medical perspective, but also a social one.

An individualized aging process is significantly influenced by preceding factors connected with the level of education, marital status, performed social functions, belonging to social groups, lifestyle, and a whole network of social correlations. In turn, aging processes lead to involutional changes that limit the resources of body organs and raise susceptibility to chronic diseases, which not only generate the necessity to use medical services, but also increase patients’ care dependency [1,2,3].

To assess the level of care dependency, a whole variety of high-quality, reliable, and up-to-date instruments are applied in clinical practice and in studies all over the world. In order to identify health problems and make correct decisions, the assessment results must be unbiased and precise [4]. Moreover, the application of a reliable and effective research tool assessing care dependency will lead to more precise clinical results, as the reliability and validity coefficients are specific for a given population and depend on the frequency of the incidence of examined features.

In geriatrics, there are numerous tools designed to assess patients’ functional status, which is the first step in assessing the multidimensional care dependency of older people. These tools are invaluable in the assessment of a particular problem and include the Katz Index of Independence in Activities of Daily Living assessing basic personal tasks of everyday life (I-ADL) designed by Lawton, which allows for assessing instrumental activities of daily living, in addition to scales measuring patients’ psychological condition focusing on their emotional status and mental capacity and scales assessing social functioning. However, none of the aforementioned scales can guarantee a thorough assessment of older people’s needs in their living environment. The Care Dependency Scale (CDS) was designed to assess the multidimensional care dependency of older people in their daily living environment, and it is recommended across Europe. The scale has also been adapted to Polish conditions [5,6,7,8]. The scale was developed following Virginia Henderson’s theory of nursing and includes both physical and psychosocial aspects, which are used to assess care dependency [9,10,11]. The results of earlier studies with the application of the CDS conducted in the Netherlands confirmed the utility of this tool in identifying whether patients are at risk of developing pressure ulcers or not, both in home and institutional care environments [12]. The connection between care dependency and other concomitant diseases such as dyspnea or incontinence was confirmed in a study carried out in Spain on a group of older patients suffering from heart failure [13]. The utility of the CDS was also demonstrated in Asia among stroke survivors receiving emergency or long-term treatment [11], as well as in Poland among geriatric ward patients suffering from exacerbation of chronic diseases or the incidence of new health problems [8] and among patients with mental disorders receiving psychogeriatric treatment within hospitals or outpatient care settings [5]. An age-related increase in care dependency among hospitalized patients was confirmed by means of the CDS in a study conducted in Germany [8].

The main aim of the present study was to determine the connection between sociodemographic factors and older patients’ care dependency in their living environment according to the CDS.

## 2. Material and Methods

The study was carried out from May to August 2018 among 151 people visiting an outpatient clinic. Before taking part in the study, each participant was informed about its purpose, method of participation, the fact that the study was anonymous, as well as about the possibility of withdrawing from participation at any stage. It was conducted by a qualified nurse (who was a member of the research team and was also responsible for measuring/assessing patients’ weight and height). The nurse read out both questions and possible answers for every participant of the survey. The questionnaire consisted of closed-ended questions. The task of each respondent was to choose an answer to a given question from a list of options. The answers were immediately written down by the qualified nurse in a questionnaire form. The survey was conducted in a separate room. Inclusion criteria for the study: patient aged over 65, ability to maintain verbal contact, stable course of the disease with which an older person came to the outpatient clinic, no severe impairment of cognitive functions (4–6 points—moderate cognitive impairment, 7–11 points—no impairment according to Abbreviated Mental Test Score–AMTS), obtaining the patient’s consent to participate in the study. Exclusion criteria: patient aged under 65, no verbal contact, unstable course of the disease with which an older person came to the outpatient clinic, severe cognitive impairment (0–3 points according to AMTS), lack of patient’s consent to participate in the study. The assessment of care dependency and the evaluation of biopsychosocial need satisfaction were conducted with the application of the CDS. The CDS measures the level of care dependency in relation to eating/drinking, continence, body posture, mobility, day and night rhythms, getting dressed and undressed, body temperature, personal hygiene, avoiding danger, communication and social contact, sense of rules and values, daily activities, recreation, and learning ability. The questions related to all of these categories were evaluated by means of a five-point Likert scale to define the level of care dependency (1—completely care-dependent; 2—high care dependency; 3—partial care dependency; 4—low care dependency; 5—almost independent of other people’s care). The total score obtained by patients was interpreted in the following way: A score between 60 and 75 accounted for a low level of care dependency, a score of 45–59 represented medium dependency, and a score of 15–44 denoted a high level of care dependency.

The internal consistency of this scale was confirmed by a high Cronbach’s alpha coefficient (0.98). Previous studies also showed that the Polish version of the CDS can be used to compare and develop international standards of assessment as far as older patients’ needs are concerned [6,8,13,14].

Personal activities of daily living (P-ADL) were assessed by means of the Katz Index of Independence in Activities of Daily Living (ADL), including questions related to independence bathing, dressing, going to toilet, transferring from bed to a chair, feeding, and continence. The scale consisted of six yes/no questions, with one point being scored for each “yes” answer (i.e., a respondent is able to perform a particular function). The total score for the whole scale is 6: A score of 5 or 6 indicates full function, a score of 3–4 represents moderate impairment, and a score of 2 or less denotes severe functional impairment in the activities of daily living [15].

Instrumental activities of daily living (I-ADL) were analyzed with the application of Lawton’s I-ADL Scale, which focuses on functions such as the ability to use the telephone, mode of transportation, food preparation, housekeeping (cleaning, house maintenance, and laundering), responsibility for one’s own medication, and ability to handle finances. The questions were accompanied by a descriptive scoring system: Performing a specific function without help (an independent person), i.e., self-reliant within a particular activity (3 points); performing a specific function with a little help from other people (2 points); inability to perform a function, dependence on others (1 point). The scores were interpreted in the following way: A score of 19–27 denoted full function, a score of 10–18 represented moderate dependency, and a score of 9 points or less indicated a high level of dependency [16].

Nutritional status was analyzed by means of the Mini Nutritional Assessment (MNA), which is a basic screening test recommended by numerous institutions for the assessment of the risk of malnutrition in people aged 65 or more. The MNA consists of four parts, which include 18 questions. The first part consists of anthropometric data: Body mass index (BMI; calculated by dividing the patient’s weight in kilograms by height in meters squared—BMI = kg/m^2^), mid-arm circumference, and calf circumference. The second part refers to data connected with the patient’s health condition, taken medicines, and everyday activity. Evaluation of the patient’s diet, which is included in the third part, consists of questions related to the quantity and quality of meals. In the last part, the respondent is asked for a self-evaluation of their nutrition and health condition. The total score of the MNA may reach 30. A score of 24 or more is interpreted as a good nutritional status, a score of 17–23.5 denotes a risk of malnutrition, and a score below 17 indicates malnourishment [17]. The 15-item Geriatric Depression Scale (GDS) is used for assessing a subjective sense of being depressed. It is a useful screening tool that makes it possible to detect an increase in depression symptoms in older people within the preceding 2 weeks. The yes/no questions refer to, for example, satisfaction with life, the respondent’s dominating mood, or sense of emptiness, and they are scored (0/1) according to a specially designed key. Scores between 0 and 5 are considered normal, scores of 6–10 indicate mild or moderate depression, and scores of 11–15 are indicative of severe depression [18].

The research was carried out in accordance with the principles of the Declaration of Helsinki.

### Statistical Analysis

The distribution of the categorical variables is presented as absolute (*n*) and relative numbers (%). The distribution of the age variables is described by the mean and standard deviation (SD), whereas the distribution of the other quantitative variables is presented as the median (Me) and the first (Q1) and third (Q3) quartiles. An analysis of the correlations among the categorical variables was conducted with application of the chi-squared test if the expected values in at least 80% of all cells of the cross table presenting a particular correlation were higher than 5. Otherwise, for 2 × 2 table sizes, Fisher’s exact test was applied, with the chi-squared exact test applied for tables of other sizes. An analysis of the correlation between the respondents’ age and the categorical variables was conducted by means of the Student’s *t*-test for independent groups, whereas the Mann–Whitney test was applied in the case of the other quantitative variables. The strength of the correlations between the qualitative variables was measured using Cramer’s V coefficient. Differences in test probability lower than 0.05 were considered to be statistically significant. Statistical analyses were conducted with IBM SPSS Statistics 24 for Windows (IBM Corp. Armonk, NY, USA).

Due to the low number of respondents, the results of certain scales were combined in the statistical analysis as follows:
−CDS: The group with a high level of care dependency was combined with those with the group with a medium level of care dependency;−ADL: The group with severe functional impairment was combined with those with the group with moderate impairment;−I-ADL: The group with high dependency was combined with those with the group with moderate dependency;−GDS: The group with severe depression was combined with those with the group with moderate depression;−MNA: The group suffering from malnourishment was combined with that at risk of malnutrition.

In the examined group of 151 older people, the percentage of examined women was higher than the percentage of examined men (85 (56.3%) vs. 66 (43.7%)). The detailed sociodemographic characteristics are presented in Table 1.

## 3. Results

The assessment of care dependency conducted with the application of the CDS showed a low level (score ranging from 60 to 75) in 84.8% (128) of the respondents, whereas a medium or high level (score between 15 and 59) was observed in 15.2% (23) of the respondents. The examined women were more likely to report a medium or high level of care dependency than the men (22.4% vs. 6.1%), while a low level of care dependency (60–75 points) was more frequent in the men than the women (93.9 vs. 77.6; *p* = 0.006).

The assessment of the personal activities of daily living showed, in the case of the respondents with full function, a low level of care dependency measured by means of the CDS compared to the respondents with a medium or high level of care dependency (96.9% vs. 3.1%; *p* < 0.001). The respondents with a high level of functional status within the I-ADL were more likely to report a low level of care dependency according to the CDS than a medium or high level (96.2% vs. 3.8%; *p* < 0.001). A connection between the self-assessed sense of being depressed according to the GDS and the assessed care dependency also proved that the respondents with a low level of care dependency were frequently characterized by a lack of depressive moods rather than a moderate sense of depression (96.9% vs. 3.1%; *p* < 0.001). The assessment of nutritional status according to the MNA showed that the respondents with a medium or high care dependency according to the CDS were more likely to develop a risk of malnutrition or were already suffering from malnourishment than those belonging to the group with a good nutritional status (91.3% vs. 8.7%; *p* < 0.001). The connections between the respondents’ assessments of functional status, sense of depression, nutritional status, and care dependency level according to the CDS are presented in Table 2.

In the group of examined men, a stronger connection was observed between the level of functional status within ADL activities and care dependency according to the CDS (V = 1) than in the group of women (V = 0.73), as well as between I-ADL and care dependency according to the CDS (V = 1) than in the group of women (V = 0.72) (Table 3).

The respondents with a medium/high level of care dependency according to the CDS were older than the respondents with a lower level of care dependency (*p* < 0.001). Married people were more likely to report a low level of care dependency according to the CDS than a medium/high level of care dependency (*p* < 0.001). The number of respondents with a medium/high level of care dependency according to the CDS decreased along with an increase in the respondents’ education level (*p* = 0.005) (Table 4).

## 4. Discussion

Following the escalating phenomenon of an aging society, the problem connected with providing older people with appropriate care is becoming a frequent and considerable challenge for all health and social care workers and providers. The unquestionable priority in the care provided to older people is to enable them to function independently in their own home for as long as possible and to improve their quality of life.

The findings of this study conducted in Poland on a group of 151 people aged over 65 and provided with outpatient care showed that care dependency measured by means of the CDS was higher in women than in men and increased along with growing functional impairment within personal and instrumental activities of daily living in both women and men. These findings are consistent with the results of the study conducted by Doroszkiewicz et al. on a group of 200 people aged over 60 and treated on a geriatric hospital ward due to exacerbation of their chronic diseases or incidence of new health problems. This study proved that care dependency measured by the CDS is closely linked to patients’ functional status within both personal and instrumental activities of daily living and increase with the patient’s age. No correlation was found between the respondents’ gender and their level of care dependency; however, the respondents with a high level of care dependency were more susceptible to impairment with respect to their cognitive functions and were more likely to suffer from pressure ulcers, falls, or depression [8]. The influence of the respondents’ gender on the level of care dependency was observed in a similar study conducted in Spain by Juárez-Vela et al. on a group of older patients diagnosed with heart failure. The study showed that women are characterized by a higher level of care dependency than men. The men belonging to a younger examined age group (average age of 77.46 ± 9.76 years) reported lower care dependency than those who belonged to the older group (average age of 83.05 ± 7.20 years). Moreover, the study proved that care dependency in men increases with their age [13]. An increase in care dependency, along with age, measured by means of the CDS was also observed in a study conducted by Lohrmann et al. on a group of 1806 patients aged over 60 and hospitalized in various German hospitals. This study proved that, with respect to the satisfaction of basic needs, the care dependency of people aged over 80 is 2–3.5-fold higher than that in younger age groups [14]. The connection between care dependency and patients’ functional status was also found in other studies, implying that higher independence in performing personal activities of daily living results in lower care dependency [19], which increases with the age of patients; however, their study did not confirm a connection between respondents’ gender, marital status, or living arrangements and their care dependency [5].

Numerous studies have shown that the life circumstances and functional status of older people depend, to a large extent, on their family status. The higher male mortality rate and the gender-related differences in life expectancy determine the marital status of older people. In the population of older people, men usually belong to the group of married respondents (78% compared to 42% of women), whereas women in the same age group are typically widowed; this trend escalates with age [20]. In this study, the authors observed that married women are characterized by a lower level of care dependency, and comparable results were obtained in another study conducted by Muszalik et al. in which a lower level of care dependency was reported by married respondents. Their study did not show a connection between respondents’ gender, place of residence, education, or socioeconomic status and care dependency; however, it showed a direct correlation between age and care dependency [21]. However, an observed between socioeconomic status and nutritional status, cognitive functions and functional status was found in the study conducted by Shahar et al. in a group of 2237 older people in Malaysia. The study showed that older people with low socioeconomic status were at a higher risk of developing health problems [22], which, in turn, speeded up the aging process [23]. These discussed relations between sociodemographic data and care dependency belonged also to the aims of the study conducted by Dueñas et al. in which widowed women living alone with a low level of education were found to be more care-dependent [24].

The connection between education and care dependency was confirmed in this study, showing that women with a higher level of education are characterized by a lower level of care dependency. Similar results were obtained by Haor et al. for a group of older people with higher education who presented a better functional status within personal and instrumental activities of daily living, along with a lower level of care dependency than people with lower education. The respondents who took advantage of services provided within primary healthcare also reported a decrease in their functional status in instrumental activities of daily living, which accompanied their advancing age [25].

Following the escalating phenomenon of an aging society, there appears a need for a far-reaching assessment of the psychosocial problems of older patients, not only to shape proper policies providing them with essential health care, but also to prepare appropriate support and nursing services. The primary aim of these activities should be to obtain information applicable for particular patients with respect to their individual circumstances. The presented findings of this study might provide directions for further research, as the topic requires further investigation and continuation. Inclusion of the CDS in everyday practice by a therapeutic team should constitute the basis of cooperation with older patients, which will make it possible to upgrade the idea of therapeutic success from a strictly biological dimension to a biopsychosocial one. However, such transformations can only be possible through further analyses in this research area, and the presented findings might provide directions for further exploration.

## 5. Conclusions

The survey results confirmed that sociodemographic factors have a significant impact on the level of care dependency. When planning care in the home environment, special attention should be paid to older women, who are more likely to lose their independence than men. These women should be given additional support. CDS scale might be applied to compare the results of the studies conducted in various countries, which may contribute to developing international standards for assessing the needs of older people. These transformations might only be possible if further analyses in this field are conducted and the findings obtained in this study should be used as directions for further exploration, which should, undoubtedly, be deepened and continued.

## Figures and Tables

**Table 1 healthcare-09-00114-t001:** Sociodemographic characteristics of the examined group.

Sociodemographic Characteristics		Women		Men	
		*N* = 85	56.3%	*N* = 66	43.7%
Age	Mean	71.5		70.7	
	SD	7.2		5.8	
	Min.	65.0		65.0	
	Max.	101.0		97.0	
	Median (Q1–Q3)	68.0 (66.0–75.0)		69.0 (67.0–73.0)	
	*p*	0.45			
Education	Primary	19	22.4	8	12.1
	Vocational	18	21.2	36	54.5
	Secondary	35	41.2	12	18.2
	Higher	13	15.3	10	15.2
	*p*	˂0.001			
Marital status	Married	32	62.4	53	80.3
	Single	32	37.6	13	19.7
	*p*	0.02			
Living arrangements	Alone	75	88.2	54	81.8
	With family	10	11.8	12	18.2
	*p*	0.26			
Socioeconomicstatus	Poor/average	46	54.1	41	62.1
	Good/very good	39	45.9	25	37.9
	*p*	0.32			

*N*—number of respondents; %—percentage of respondents; SD—standard deviation; Min.—minimum; Max.—maximum; Q1—lower quartile; Q3—upper quartile; *p*—*p*-value for Student’s *t*-test for independent samples; *p*—*p*-value for Pearson’s chi-squared test/Fisher’s exact test.

**Table 2 healthcare-09-00114-t002:** Functional status, sense of depression, nutritional status, and care dependency according to the Care Dependency Scale (CDS) in the examined groups of women and men.

Scales	Functional Status	Women *N* (%)		Men *N* (%)	
		Level of Care Dependency According to CDS			
		Low (60–75 Points)	Medium/High (15–59 Points)	Low (60–75 Points)	Medium/High (15–59 Points)
ADL	Full function (5–6 points)	62 (93.9)	4 (6.1)	62 (100)	0 (0.0)
	Moderate/severe impairment (<5 points)	4 (21.1)	15 (78.9)	0 (0.0)	4 (100)
	*p*	<0.001		<0.001	
I-ADL	High (19–27 points)	63 (92.6)	5 (7.4)	62 (100)	0 (0.0)
	Medium/low (<19 points)	3 (17.6)	14 (82.4)	0 (0.0)	4 (100)
	*p*	<0.001		<0.001	
GDS	No sense of depression (0–5 points)	62 (87.3)	9 (12.7)	62 (93.9)	4 (6.1)
	Mild depression/severe depression (6–15 points)	4 (28.6)	10 (71.4)	0 (0.0)	0 (0.0)
	*p*	<0.001		0	
MNA	Good nutritional status (≥24 points)	44 (97.8)	1 (2.2)	54 (98.2)	1 (1.8)
	Risk of malnutrition/malnourishment (<24 points)	22 (55)	18 (45)	8 (72.7)	3 (27.3)
	*p*	<0.001		0.01	

*N*—number of respondents; *p*—*p*-value for Fisher’s exact test; ADL—Personal Activities of Daily Living; I-ADL—Instrumental Activities of Daily Living; GDS—Geriatric Depression Scale; MNA—Mini Nutritional Assessment (MNA).

**Table 3 healthcare-09-00114-t003:** Connection between the functional status of the examined women and men assessed by means of ADL and I-ADL scales and their level of care dependency according to the CDS.

Scale	Gender	CDS	
		*p*	V
ADL	Women	<0.001	0.73
	Men	<0.001	1
I-ADL	Women	<0.001	0.72
	Men	<0.001	1

*p*—*p*-value for Fisher’s exact test; V—value of Cramer’s V coefficient.

**Table 4 healthcare-09-00114-t004:** The sociodemographic characteristics and level of care dependency in women and men according to the CDS.

Sociodemographic Characteristics		Women		Men	
		Care Dependency Level According to CDS			
		Low (60–75)	Medium/High (15–59)	Low (60–75)	Medium/High (15–59)
Age	Mean	69.6	78.2	69.9	82.5
	SD	5.1	9.4	4.6	10.0
	Min.	65	65	65	74
	Max.	86	101	85	97
	Median (Q1–Q3)	68.0 (66.0–72.0)	81.0 (70.0–83.0)	69.0 (67.0–71.0)	80.0 (77.0–89.0)
	*p*	0.001		0.087	
Education	Primary	9 (47.4)	10 (52.6)	8 (100)	0 0
	Vocational	16 (88.9)	2 (11.1)	34 (94.4)	2 (5.6)
	Secondary	29 (82.9)	6 (17.1)	12 (100)	8 (80)
	Higher	12 (92.3)	1 (7.7)	0 0	2 (20)
	*p*	0.003		0.19	
Marital status	Married *N* (%)	48 (90.6)	5 (9.4)	50 (94.3)	3 (5.7)
	Single *N* (%)	18 (56.3)	14 (43.8)	12 (92.3)	1 (7.7)
	*p*	<0.001		1.0	
Living arrangements	Alone	58 (77.3)	17 (22.7)	51 (94.4)	3 (5.6)
	With family	8 (80.0)	2 (20.0)	11 (91.7)	1 (8.3)
	*p*	1.000		1.000	
Socioeconomic status	Low/average	34 (73.9)	12 (26.1)	38 (92.7)	3 (7.3)
	Good/very good	32 (82.1)	7 (17.9)	24 (96.0)	1 (4.0)
	*p*	0.37		0.6	

SD—standard deviation; Min.—minimum; Max.—maximum; Q1—lower quartile; Q3—upper quartile; *p*—*p*-value for Student’s *t*-test for independent samples.

## Data Availability

The data presented in this study are available on request from the corresponding author. The data are not publicly available due to ethical guideline.

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
