# Peer review of "Sociodemographic Factors Affecting Older People’s Care Dependency in Their Daily Living Environment According to Care Dependency Scale (CDS)"

_healthcare, 2021, doi:10.3390/healthcare9020114_

Round 1
Reviewer 1 Report
Congratulations, it´s an interesting research.
I have a question.
It´s about in relation to: "Nutritional Status" and "Financial Status" (Socioeconomic Status).
I present you these articles and I would like your comment in relation to your research.
1.- Suzana Sahara , Divya Vanoh, Tengku Aizan Hamid
Factors associated with poor socioeconomic status among Malaysian older adults: an analysis according to urban and rural settings.
BMC Public Health 19 Article number:549 (2019).
2.- Andrew Steptoe and Paola Zaninotto.
Lower Socioeconomic Status and the acceleration of aging: An outcome-wide analysis.
PNAS June 30, 2020 117(26) 14911- 14917
And may be , you can do a comment about these papers in the discussion.
Thank you in advance.
Author Response
Dear Editor,
Dear Reviewers,
Thank you very much for all your comments and suggestions. I am also very grateful for the articles which you have sent to me as they helped me enrich the discussion of my manuscript.
In the table 1 end 4 “Financial status” inserted „ Socioeconomic”
Best regards
Grażyna Puto
Reviewer 2 Report
Summary in the abstract and conclusion are not well aligned. Abstract does not emphasize gender difference, yet conclusion does. The statement about Education in conclusion is not clear and different than the abstract.
The primary flaw of this manuscript is the absence of detail related to participant recruitment and the data collection which in turn challenges the interpretation of the results. To address this deficit, the following needs to be described: recruitment methods, circumstances of data collection (household? clinic based?), were participants aided in completing questionnaires, were data collectors trained to collect data verbally, process of gaining consent and/or were height/weight self-report? Were accommodations made for participants having low literacy and/or visual impairment?
The conclusions are quite brief and do not offer a significant contribution to the current guidance for assessment of elderly patients' level of care dependency.
Author Response
Dear Editor,
Dear Reviewers,
Thank you very much for all your comments and suggestions.
The summary and conclusions have been expanded taking into account older people’s care dependency. The discrepancies which appeared between the results in the abstract and conclusions have been removed.
Materials and methods included in the manuscript were significantly expanded by explaining recruitment methods, inclusion and exclusion criteria for the participants of the study and description of the circumstances of collecting data. The conclusions have been expanded taking into account older patients’ care dependency.
Best regards.
Results: The findings show that 84.8% of the respondents reported a low level of care dependency, whereas 15.2% were characterized by a medium or high level. The respondents with a medium or high level of care dependency were older than those with a low level (p < 0.001). The respondents with a low level of care dependency were more likely to be married than the respondents with a medium or high dependency level (p < 0.001). The number of people with a medium or high level of care dependency decreased along with an increase in the respondents’ level of education (p = 0.005).
Results: Gender had a significant impact on the level of care dependency. The surveyed females obtained the medium or high level of dependency more often than males (22.4% vs 6.1%), and the low level of dependency was significantly more frequent among men than women (p = 0.006). Also the age of the respondents determined their level of care dependency. The subjects with a medium or high level of care dependency were significantly older (p = 0.001).The subjects with a low level of care dependency were more likely to be married than people with a medium / high level (p <0.001).The level of education had a significant impact on care dependency. A higher level of education correlated with a medium / high level of dependency (p = 0.003)
Conclusions: The aspects that should be taken into consideration in order to determine elderly patients’ care dependency in their daily living environment include gender, age, level of education, and marital status.
Conclusions: The survey results confirmed that sociodemographic factors have a significant impact on the level of care dependency. When planning care in the home environment, special attention should be paid to older women, who are more likely to lose their independence than men. These women should be given additional support.
Material and Methods
The study was carried out between May and August 2018 in a group of 151 elderly patients (aged over 65, with maintained verbal contact, examined in a period of disease stability, without paresis, oncological diseases, or disorders of cognitive functions—4–11 points on Abbreviated Mental Test Score (AMTS), staying in their place of residence, receiving treatment in an outpatient clinic, and having given their informed consent for participation in the study). The questionnaire consisted of questions concerning sociodemographic characteristics and standardized scales.
Material and Methods
The study was carried out from May to August 2018 among 151 people visiting an outpatient clinic. Before taking part in the study, each participant was informed about its purpose, method of participation, the fact that the study was anonymous, as well as about the possibility of withdrawing from participation at any stage. The interview was conducted in a separate room. The study was conducted using a questionnaire that was distributed by a qualified nurse (who was a member of the research team). The questionnaire consisted of closed-ended questions. The task of each respondent was to choose an answer from a list of destructors assigned to a given question. Inclusion criteria for the study: patient aged over 65, ability to maintain verbal contact, stable course of the disease with which an older person came to the outpatient clinic, no severe impairment of cognitive functions (4- 6 points – moderate cognitive impairment, 7 – 11 points – no impairment according to Abbreviated Mental Test Score -AMTS), obtaining the patient's consent to participate in the study. Exclusion criteria: patient aged under 65, no verbal contact, unstable course of the disease with which an older person came to the outpatient clinic, severe cognitive impairment (0-3 points according to AMTS), lack of patient’s consent to participate in the study.
Reviewer 3 Report
Thank you for the opportunity to read this interesting and timely manuscript on care dependency in older people. The findings, while expected, contribute to the literature due to the importance of understanding these relationships in order to provide appropriate care for older people in society.
I have the following comments to improve the manuscript before publication:
Title: Please avoid the use of the word “elderly” throughout the manuscript. “Older people” is the more appropriate language to use, or “Older adults”. Many people over 65 years of age continue to work and may not want to be referred to as elderly.
Line 25: Please clarify “place of residence”. Do you mean living in the community or community-dwelling?
Line 25: Please replace “such as” with “including”.
Line 71: “Proven” is not appropriate here. Consider “demonstrated”.
Line 77: Do you mean to determine if there is a correlation in your population?
Page 82: For readers not familiar with the AMTS, please clarify the level of cognitive function for people who score 4-11 points.
Page 128: Please clarify whether this was the GDS Short Form.
Page 137: I am confused by the use of the word “qualitative” in section 2.1 considering you have used quantitative methods.
Line 157: Should be GDS
Line 230: It is not possible to confirm something in a cross-sectional study. Please use associated (or similar).
Line 236: Same as the previous comment.
Line 240: Same again.
Line 253: Same again.
Line 257: Same again – particularly due to low sample size which is highlighted by the combining of categories described in the methods. In addition, given this, it is surprising that there is not at least a brief description of the limitations of this study.
Author Response
Dear Editor,
Dear Reviewers,
Thank you very much for all your comments and suggestions. I do apologize for using the expression “elderly patients”, of course “older people” is a more appropriate one. I am very grateful for your suggestions as far as English edition and style are concerned. The conclusions have been expanded taking into account older patients’ care dependency.
Best regards.
Title: replaced „elderly patients” with „older people”
Line 25: replaced „place of residence” with own homes
Line 25: replaced „such as” with „including”.
Line 71: replaced „Proven” with „demonstrated”
Line 77: replaced „correlation” with „connection”
Page 82: Material and Methods information provided „(4- 6 points – moderate cognitive impairment, 7 – 11 points – no impairment according to Abbreviated Mental Test Score -AMTS)”, and „(0-3 points according to AMTS)”
Page 128: 15-item Geriatric Depression Scale
Page 137: replaced “quantitative” with „categorical”
Line 157: replaced „GSD” with „GDS”
Line 230: replaced „confirmed” with “observed”
Line 236: replaced “confirmed” with „observed”
Line 240: replaced “confirmed” with „found”
Line 253: replaced “confirmed” with „showed”
Line 257: replaced „correlation” with „connection”
Round 2
Reviewer 2 Report
Revisions to the Methods and Conclusion are much improved. However, the methods section needs more detail - 1) still not clear if the interview was verbal or written, if written were accommodations made for low literacy or visual impairment (given the age group)? ; 2) if the questions were close ended, this method seems to be questionnaire administration not interview; and 3) odd placement of the term destructors.
Conclusion still seems quite brief. Can any statements be offered in relation to administering a questionnaire to this age group?
Author Response
Dear Editor,
Dear Reviewers,
Thank you very much for all your comments and suggestions.
In the material and methods section the description of the conducted research was developed and some details were added.
The conclusions were extended by adding the information about a possibility od applying CDS in international studies.
Line 85: Material and Methods - description added
Line 94: “destructors” - word removed
Line 298: The conclusions were extended